

# Evaluating the stress-response of dental students to the dental school environment

Katarzyna Mocny-Pachońska[1,*], Rafał Doniec[2,*], Agata Trzcionka[1], Marek Pachoński[3], Natalia Piaseczna[2], Szymon Sieciński[2], Oleksandra Osadcha[4], Patrycja Łanowy[1] and Marta Tanasiewicz[1]

[1] Department of Conservative Dentistry with Endodontics, Medical University of Silesia, Faculty of Medical Science, Bytom, Poland
[2] Faculty of Biomedical Engineering, Department of Biosensors and Biomedical Signal Processing, Silesian University of Technology, Zabrze, Poland
[3] Pachonscy Dental Clinic, Strzebiń, Poland
[4] Silesian University of Technology, Institute of Mathematics, Gliwice, Poland
[*] These authors contributed equally to this work.

Corresponding author
Katarzyna Mocny-Pachońska,
kpachonska@sum.edu.pl

## ABSTRACT

**Introduction and Objective**. Dentists experience high amounts of professional stress beginning with their student years in dental school. This stress, given its early onset, may negatively impact the personal and professional lives of these individuals, as well as the quality of their clinical work. We sought to create an objective scale to evaluate the levels of stress in students at different stages of their education, as well as in practicing physicians.

**Materials and Methods**. Thirty dental students participated in this study, with 10 students each selected from junior, mid-senior, and senior classes. They were randomly divided into two groups in which one group was subjected to stressors while the other group was not. JINS MEME ES_R (JINS) smart glasses and Garmin Vivoactive 3 smartwatches were used to obtain data, including electrooculography (EOG), heart rate (HR), and accelerometer (ACC) and gyroscope (GYRO) feedback, while the subjects performed a dental exercise on a phantom tooth.

**Results**. The heart rates of more experienced students were lower than those of the junior students. The EOG, ACC, and GYRO signals showed multiple differences in the measurement of amplitudes and frequency of episodes.

**Conclusion**. Our pilot results show that electronic tools, like smart glasses with software and sensors, are useful for monitoring the stress levels of dental students in preclinical operating conditions. We would like to further assess the stress levels in students performing dental procedures on phantom teeth and in later clinical interactions with patients.

## INTRODUCTION

Dentists experience high levels of professional stress which can negatively affect them from their student years working with phantom teeth through their preclinical and clinical practice years (*Al Faris et al., 2016*).

Stress related to dental school can negatively impact a dentist's personal and professional life, particularly with regard to the quality of their clinical work (*Rada & Johnson-Leong, 2004*; *Piazza-Waggoner et al., 2003*; *Harazin et al., 2014*; *Farokh-Gisour & Hatamvand, 2018*; *Chandrasekaran, Cugati & Kumaresan, 2014*). Stress resulting from external physical or mental factors can affect an individual's physical and psychological well-being and may reduce work efficiency and impact the accuracy of diagnostic and therapeutic decisions (*Rada & Johnson-Leong, 2004*; *Harazin et al., 2014*).

It is difficult to predict and control reactions to external stressogenic situations. During stressful conditions, the hypothalamus-pituitary-adrenal (HPA) axis and the sympathetic nervous system (SNS) are activated. The nervous system becomes aroused in periods of stress or danger, which affects an individual's reactions. Responses to stressful situations vary among individuals due to diverse perceptions of circumstances. The stress response is more highly aroused when a stressor is directed in a personal manner and continues with the ongoing risk of threat (*Martinez et al., 2017*; *Schachner & Singer, 2000*).

Studies have shown that students from medical universities are more prone to psychological problems due to unique stressors, including the demands of their education and special expectations compared to students of humanities and technical faculties (*Piazza-Waggoner et al., 2003*). *Dyrye, Thomas & Shanafelt (2006)*. reported that medical students experience high levels of anxiety and depression. *Mohannad et al. (2019)* and *Brazeau et al. (2014)* confirmed that depression in dental students and interns appears more frequently than in the general population, with evidence of mental health deterioration over the course of their medical training. *Al Faris et al. (2016)* emphasized that the third and fifth years of dental and medical studies are highly associated with depressive symptoms in students.

Dental school is a significant source of stress for dental students who are required to practice in a professional manual workshop for clinical work, establish direct patient contact, and manage patient stress. These stressors are in addition to their rigorous educational challenges, which include providing care for patients and performing treatments that could cause harm if performed improperly.

Dental students must deal with life-threatening conditions, work long hours, undertake a demanding workload, and master an intense theoretical education (*Harazin et al., 2014*). Studies show that young dental students can significantly improve complicated fine motor skills, such as performing dental work while looking in a mirror, with practice (*Lugassy et al., 2019*).

Dental students appear to be more at-risk for stress than students from other faculties, including medical students, because the need to acquire extensive knowledge is combined with manual training, which is practiced in a preclinical simulation center and then in clinical practice (*Ali et al., 2018*).

When training in clinical practice, students have direct contact with patients who may be in pain and fearful of dental examinations and procedures (*Ali et al., 2018*). Our findings support the results demonstrated by *Abu-Ghazaleh, Sonbol & Rajab (2016)* that stress in dental students at the University of Jordan increased over the years of the clinical training; fifth-year students showed especially high levels of psychological stress due to an increased awareness of the expectations and responsibilities of their work. The negative impact of depression on dental students is replaced by stress resulting from the need to cope with professional duties and full-time clinical work, and the need to simultaneously carry out continuing postgraduate education (*Roberts, 2010*).

Stress adversely affects the quality of patient care, patient safety, and professionalism, and its effects extend to the community at large (*Al Faris et al., 2016*). Therefore, given the high rate of depressive symptoms in dental students, stress levels and their causative factors should be determined and preventive measures should be considered (*Davidovich et al., 2015*).

*Davidovich et al.*'s (*2015*) research comparing students and experienced professionals shows that stress levels are significantly higher among newcomers to clinical practice. Junior students rely on theoretical knowledge from preclinical classes, which may not yet be sufficient, and are typically anxious about using dental equipment with precision (*Davidovich et al., 2015*).

*Piazza-Waggoner et al. (2003)* and *Davidovich et al. (2015)* investigations confirmed that dental students practicing on patients in a clinical setting have a particularly high level of stress during the implementation of procedures related to invasive work on patients, especially when practicing on a child.

The environment in which students perform their clinical work is also significant. The possibility of being able to perform tasks in silence, under time constraints while being able to complete the task satisfactorily, and the knowledge of how to deal with stressful situations are factors that influence a student's quality of work (*Kiesre & Herbison, 2000*; *Elani et al., 2014*).

Several recent studies have investigated the effectiveness of interventions such as psychoeducational interventions, music therapy, and Progressive Muscle Relaxation (PMR) to cope with student anxiety or stress before academic exams or evaluations (*Labrague et al., 2017*; *Ince & Cevik, 2014*; *Hashim & Zainol, 2015*).

Researchers have not yet confirmed that relaxation or stress management techniques are effective in reducing the fear of working in a clinical setting (*Davidovich et al., 2015*). It is difficult to accurately identify a person's cognitive state but it is important to have an objective real time evaluation for dentists because their cognitive state can influence their performance.

Researchers typically rely on psychological measurements to collect data (*Ishimaru et al., 2015*; *Ogawa, Takahashi & Kawashima, 2016*). However, future research should determine whether differences in the prevalence of severe depressive symptoms among students in different disciplines are due to the innate nature of the discipline, the type of curriculum being studied, or the intellectual and manual skills required (*Al Faris et al., 2016*).

## Objectives

We conducted an arbitrary, measurable study of the arousal level of 30 dental students in their third, fourth, and fifth years of study. Results were measured by blink occurrence, speed, and strength; vertical and horizontal eye movement; head movement (rotation roll, pitch, and yaw), and changes in heart rate measured by JINS smart glasses and a GPS smartwatch, with and without stress stimulation. We created our own protocol for stress simulation based on literature data (*Dedovic et al., 2005*; *Ghazali et al., 2018*). The first measured stressor was shaping a cavity in a tooth in a limited time under the careful supervision of a teacher. The second stressor was creating the shape of a cavity similar to the matrix of the cavity. The third stressor was a lack of information on how to prepare the cavity. The control group of students without the stress simulation protocol underwent a session of muscle relaxation, music therapy, and were given full information about the task to be performed. We sought to standardized the measurement method that could be used in further studies using JINS smart glasses.

## MATERIAL AND METHODS

### Technology used

We used JINS MEME ES_R (JINS) smart glasses with three-point electrooculography (EOG) and six-axis sensors(an accelerometer and a gyroscope) to detect the following actions: blink occurrence, speed, and strength; vertical and horizontal eye movement; and head movement (rotation roll, pitch, and yaw) (*Chen et al., 2012*).

Participant's heart rates (HR) were measured using the Garmin Vivoactive 3 GPS smartwatch with a heart rate sensor located on the back of the device, which displayed the current heart rate in beats per minute (bpm). A graph of the heart rate was also displayed on the device.

### Research group

The research participants were third-, fourth-, and fifth-year students (junior, mid-senior, and senior, respectively) of the Dental Division of Silesian Medical University, Katowice, Poland. Ten students were selected from each year group and randomly classified into two equinumerous subgroups: a Management Anxiety Group (MAG) and an Uncontrolled Anxiety Group (UAG). Participation in the study program was determined by the schedule of didactic classes that coincided with the start date of the experiment. Study subgroups were randomly assigned from a closed pool of tickets marked MAG or UAG.

The MAG received a session of PMR and music therapy, detailed and precise information about the task to be performed (analysis of the 3D model, and measurements of the range, shape, and depth of the tooth's cavity, both without any time limit), and recommendations to fully concentrate on precision shape modeling. Relaxation procedures were guided by an experienced physiotherapist (*Gallego-Gómez et al., 2020*): subjects were placed in a room with dimmed lights and sat in chairs to relax the main muscle groups and relieve tension. Relaxing music was also played from a portable audio system using a CD played at a volume of 50–60 DB.

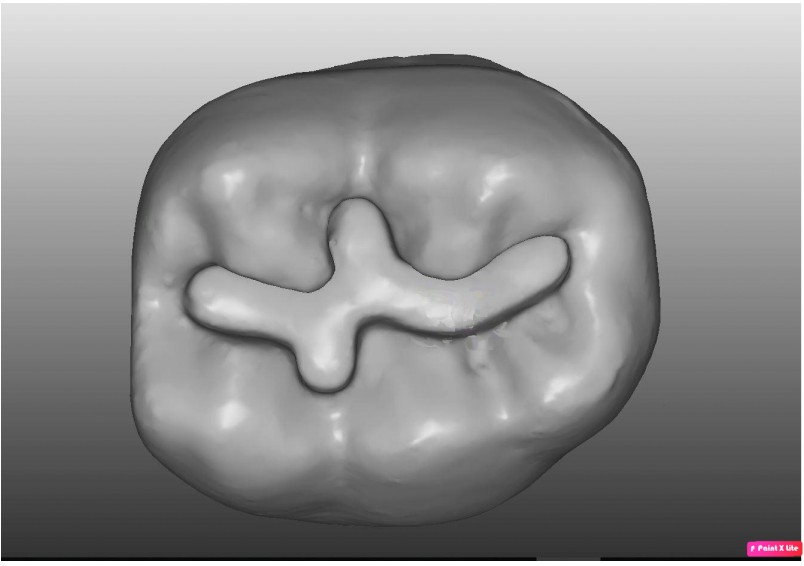

**Figure 1**    **Matrix with outline of cavity.**

The UAG was not given any information or recommendations and did not undergo relaxation training. The UAG was also given a time limit to prepare the cavity. The procedure was conducted under careful teacher supervision but without assistance. The teacher was tasked with informing the students of the passing time at 10 s intervals as well as evaluating their work using the phrase: 'It's bad, not good, hurry up' (*Dedovic et al., 2005*; *Ghazali et al., 2018*)

All participants were equipped with a set of monitoring devices (the smart glasses and smartwatch) during the cavity preparation procedure to measure the vital signs that defined their stress intensity.

## Experiment

Students were asked to perform the preparation of a cavity in a standard phantom molar, tooth 36, of a 1:1 scale (Frasaco GmbH, Germany) with limited or unlimited time, depending on the group they were assigned to. The matrix with the outline of the cavity was prepared beforehand (Fig. 1).

They used a Round End Taper NTI Rotary Dental Instruments burr with head length, four mm; diameter, 009 mm; grit, medium.

The participants performed the procedure with high speed ending (350.000 revolutions per minute) and water cooling. A pattern of the cavity outline was presented on a model (Fig. 2) prepared using a 3D printer (Formlabs 2). The model was characterized by the following features: magnification ×10, material: Photopolymer Resin Black FLGPBK04, volume: 389.82 ml, layer thickness: 0.025 mm, number of layers: 3312, print time: 57 h 13 min.

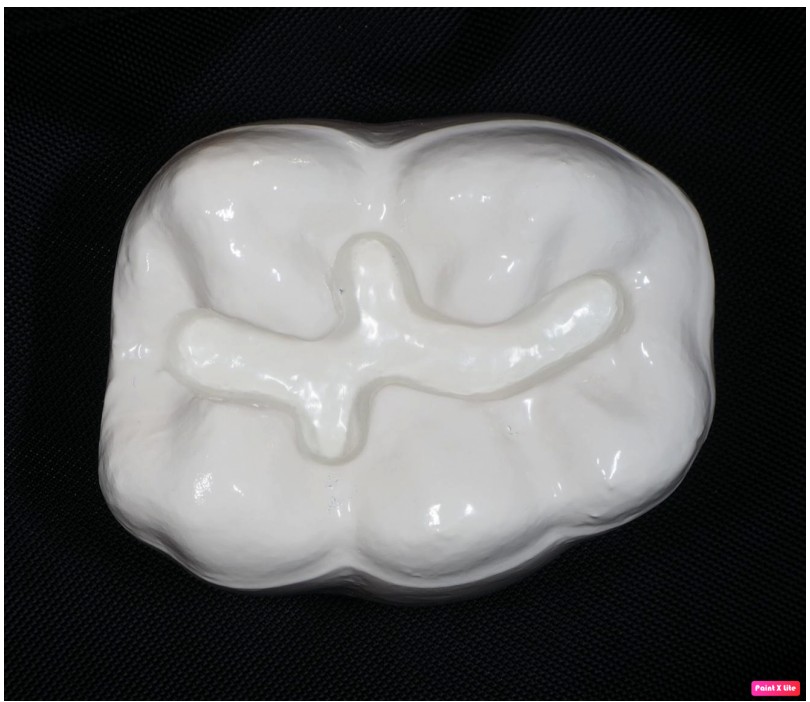

**Figure 2** Printed tooth 36 with matrix.

After a one-week period, the participants switched experimental groups and performed the same procedure, allowing us to determine the differences between MAG and UAG data for the same individual.

Data acquired from the smart glasses worn during the experiment were divided into three parts:

I.   electrooculography (EOG),

II   acceleration (ACC),

III. gyroscope (GYRO).

The data collected during the first four seconds of the experiment (when the participant started the procedure) and the last four seconds (when the procedure was completed) were not used for analysis, as the largest change in values were observed during that time.

60 experiments were conducted and one common episode that caused rapid rises of amplitude in a signal was chosen to properly evaluate the results. Consent for the experiments was issued by the Ethical Commission of Medical University of Silesia with resolution number KNW/0022/KB1/79/18 on October 16, 2018. All participants gave their written consent prior to the start of the study.

## Statistical procedures

Statistical analysis was performed using Statistica Version 9.0 (MUS Katowice, Poland). Results are presented as mean and standard deviation (SD). Statistical analysis of HR values revealed that all distributions were normally distributed. Differences among groups were analyzed using the Student's $t$-test. A Shapiro–Wilk test of normality was performed. A

**Table 1** Mean, standard deviation (SD), median and IQR of heart rate measurements in all participants.

| | | | Mean | SD | Median | IQR | *p*-value before/after *t*-student test |
|---|---|---|---|---|---|---|---|
| Junior (III) | UAG | before | 100.6 | 14.66 | 101.5 | 24 | *p* = 0.36 |
| | | after | 97.2 | 13.91 | 95.5 | 18 | |
| | MAG | before | 86.0 | 7.76 | 84 | 6 | *p* = 0.17 |
| | | after | 82.8 | 13.91 | 82.5 | 13 | |
| Mid-senior (IV) | UAG | before | 92.7 | 11.98 | 93 | 20 | *p* = 0.02 |
| | | after | 80.8 | 10.56 | 83 | 16 | |
| | MAG | before | 84.5 | 13.62 | 83.5 | 20 | *p* = 0.30 |
| | | after | 77.4 | 8.85 | 78.5 | 12 | |
| Senior (V) | UAG | before | 86.0 | 6.48 | 87 | 9 | *p* = 0.14 |
| | | after | 95.4 | 18.28 | 96.5 | 29 | |
| | MAG | before | 83.5 | 4.06 | 83.5 | 6 | *p* = 0.26 |
| | | after | 85.9 | 6.23 | 88.5 | 12 | |

one-way analysis of variance was used to compare the groups. Fisher's LSD variant was used for two-tailed post hoc tests in justified cases of this test (two cases).

# RESULTS

## HR data

The Student's $t$-test was used to evaluate the differences in the before and after results from the UAG and MAG groups. Statistically significant differences were only found in the mid-senior UAG ($p = 0.02$) (Table 1). The heart rate in this group gradually decreased while performing the dental task, which implies that stress management training is useful for less experienced students.

The ANOVA test was used to compare HR values among the UAG and it revealed significant differences between the year groups (Fig. 3). The post hoc test results show that significant differences occurred between the third- and fourth-year groups (Table 2).

The results of the ANOVA test in the UAG show significant differences in HR between year groups. The post hoc test results show significant differences when comparing juniors with mid-seniors, and mid-seniors and seniors with mid-seniors. Mid-year seniors had a lower HR than juniors and seniors. There is no statistically significant difference in HR between juniors and seniors (Fig. 4, Table 3).

No statistically significant results were obtained for the MAG (MAG before: $p = 0.69$, MAG after: $p = 0.13$). The anti-stress training indicates that HR values did not differ significantly.

## Data from EOG

Data were collected on blinks, squinting of eyes, and probable change of sight directions. There were higher amplitudes and more frequent episodes demonstrated by seven
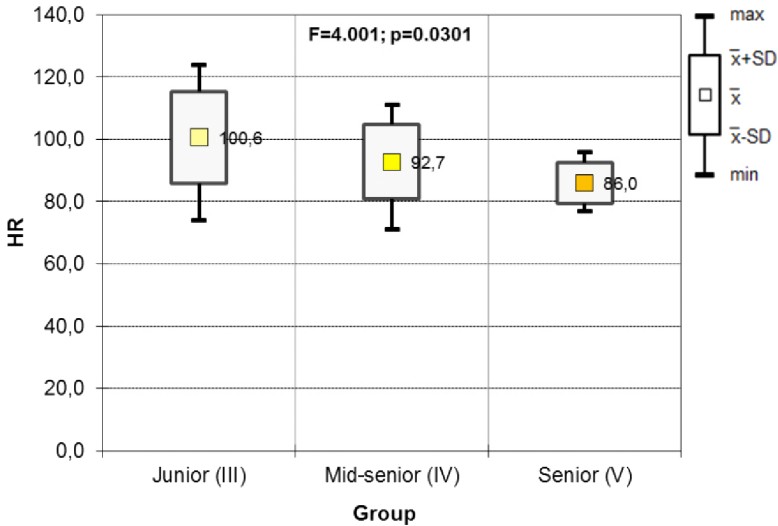

**Figure 3** Results of one-way ANOVA for HR including the year of study in UAG groups (before).

**Table 2** Results of two-tailed post hoc test (LSD Fisher) in UAG groups (before).

|  | Junior (III) | Mid-senior (IV) | Senior (V) |
|---|---|---|---|
| Junior (III) |  | 0.01 | **0.008** |
| Mid-senior (IV) | 0.01 |  | 0.20 |
| Senior (V) | **0.008** | 0.20 |  |

participants in the UAG (Fig. 5), while their counterparts in the MAG received lower scores (Fig. 6). There were more frequent blinks and squints, and higher muscle tension in eyes in the UAG.

The number of episodes of saccades and blinks was determined as the number of local maxima of the moving average of the EOG_H signal. EOG_H was used to find these episodes because, according to the producers of the JINS glasses, the greatest changes in EOG while blinking occur in the horizontal component of this signal. The signal processing steps are as follows:

1. Zero-phase 3rd order band-pass Butterworth filter (frequency range 1–35 Hz). Filtration was performed in MATLAB R2019b using the filtfilt function. The frequency range of the band-pass filter was chosen based on (*Nor'aini, Raveendran & Selvanathan, 2007*).
2. Median filtration using 5th order median filter.
3. Moving average FIR (finite impulse response) filter (window-width of 45 samples).
4. Finding local maxima using findpeaks function (Fig. 7).

Based on the comparison of the mean number of peaks, mid-senior and senior MAG students had a lower number of peaks than students from the UAG. The MAG had a higher number of peaks than the UAG in junior-level students (Table 4).

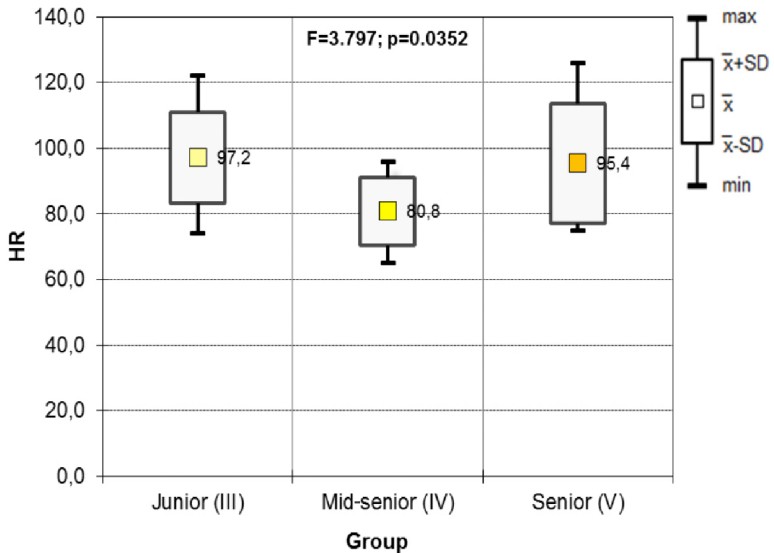

**Figure 4  Results of one-way ANOVA for HR including the year of study in UAG groups (after).**

**Table 3  Results of two-tailed post hoc test (LSD Fisher) in UAG groups (after).**

|                  | Junior (III) | Mid-senior (IV) | Senior (V) |
|------------------|--------------|-----------------|------------|
| Junior (III)     |              | **0.01**        | 0.07       |
| Mid-senior (IV)  | **0.01**     |                 | **0.03**   |
| Senior (V)       | 0.07         | **0.03**        |            |

## Data from ACC and GYRO

Data from the gyroscope were observed as tilts of the head along each plane of the body. Accelerometer data were collected from movements of the head, and their acceleration and intensity. The complete information about the dynamics of the head movements was acquired from both the accelerometer and gyroscope.

The average acceleration (ACC) of juniors and seniors in the UAG and MAG on the $x$- and $y$-axes was higher in the UAG than in the MAG, but was lower on the $z$-axis. The amplitude of the gyration on $x$-axis head tilts were larger for all students (junior, mid-senior, and senior) in the UAG than for the MAG.

The standard deviations of the head rotation angles were high when considering the average (Table 5).

The ACC_X signal was used in further analyses because of the differences in amplitude among all study groups (Figs. 8–9).

Each of the students who took part in the examination was added to the histogram. Three ranges were proposed from minimum to maximum ACC_X, with the arousal flagged. The mean values of ACC_X signal amplitudes were calculated for each of the students from the UAG and MAG in order to compare his/her stress management abilities (Table 6).

The following ranges were determined based on the data:
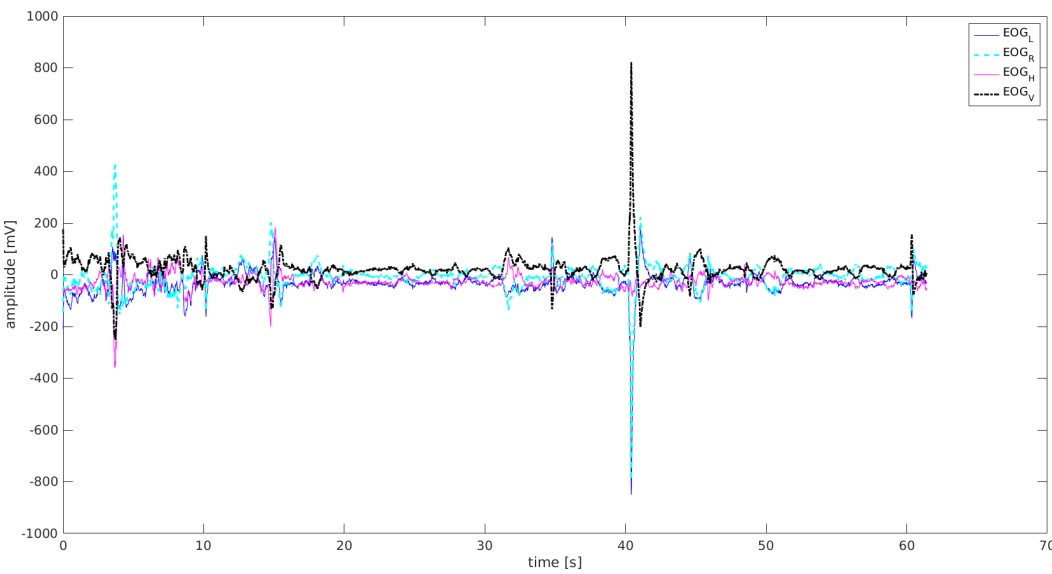

**Figure 5 EOG signal-Mid-senior student UAG.** EOGL, left eye movement; EOGR, right eyemovement; EOGV, vertical eye movement; EOGH, horizontal eye movement.

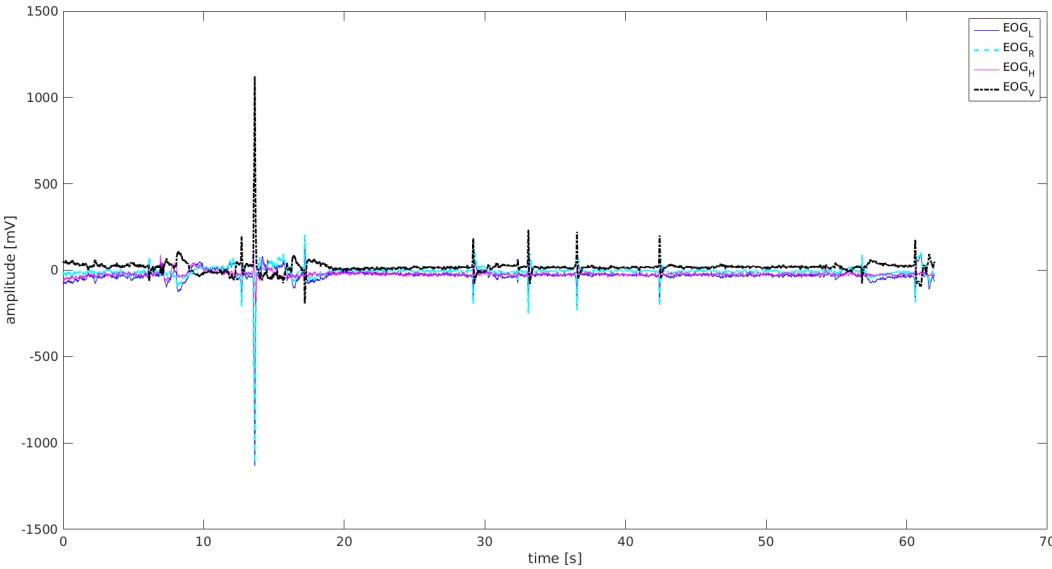

**Figure 6 EOG signal-Mid-senior student MAG.** EOGL, left eye movement; EOGR, right eye movement; EOGV, vertical eye movement; EOGH, horizontal eye movement.

1. −1509 to −726: low level of arousal;
2. −726 to 57.7: normal level of arousal;
3. 57.7 to 841: high level of arousal;

Students were classified as having a particular range of arousal based on their medium values. The greatest number of students were characterized by normal arousal levels in

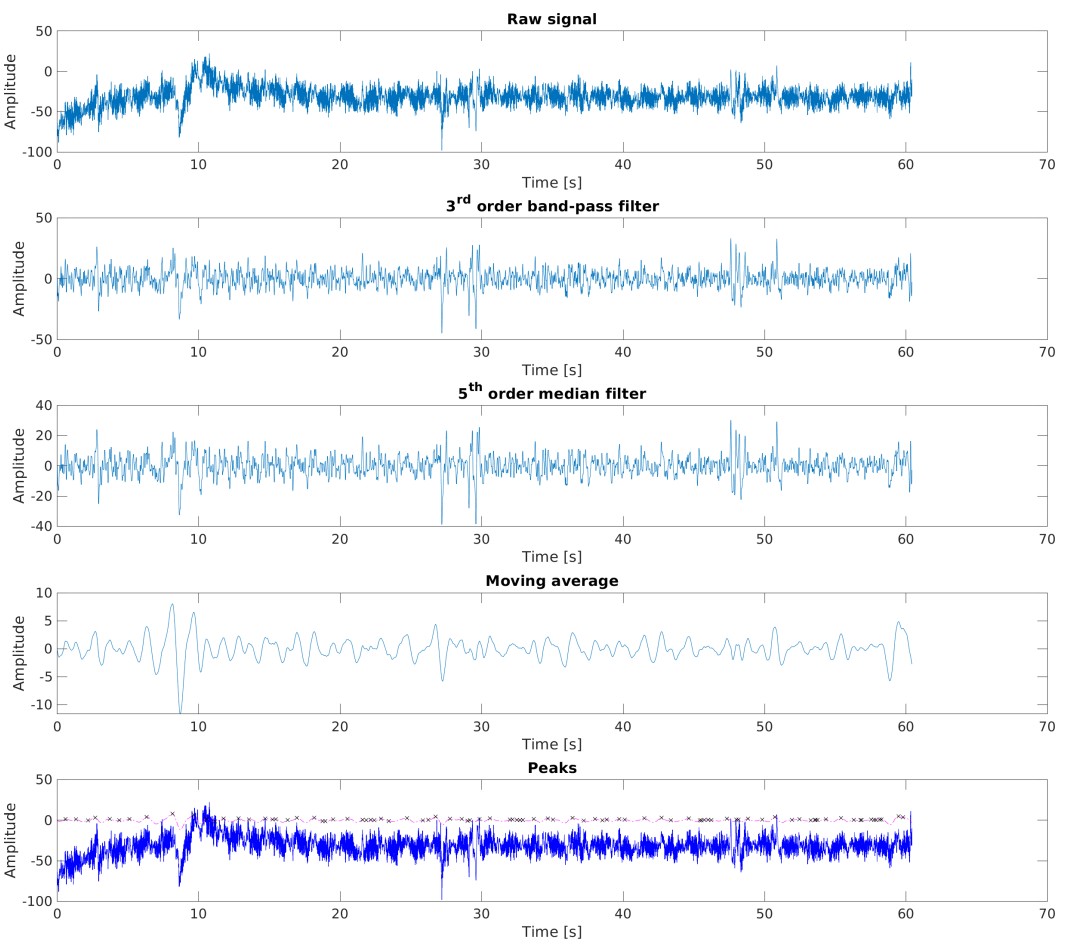

**Figure 7** **The steps of processing EOG_H signal.** EOG_H Raw signal. EOG_H 3rd order band-pass filter. EOG_H 5th order median filter. EOG_H Moving average. EOG_H Peaks.

**Table 4  Number of peaks in EOGH_signal.**

| Participant | 1 | 2 | 3 | 4 | 5 | 6 | 7 | 8 | 9 | 10 | Mean | SD |
|---|---|---|---|---|---|---|---|---|---|---|---|---|
| Junior UAG | 75 | 40 | 112 | 62 | 71 | 72 | 58 | 76 | 76 | 66 | 71 | 17 |
| Junior MAG | 87 | 92 | 80 | 71 | 95 | 69 | 58 | 102 | 100 | 59 | 81 | 16 |
| Mid senior UAG | 89 | 77 | 69 | 85 | 103 | 69 | 86 | 71 | 75 | 59 | 78 | 12 |
| Mid senior MAG | 63 | 62 | 70 | 106 | 89 | 60 | 79 | 61 | 87 | 62 | 74 | 15 |
| Senior UAG | 73 | 14 | 81 | 47 | 68 | 62 | 105 | 3 | 3 | 91 | 55 | 35 |
| Senior MAG | 8 | 16 | 40 | 74 | 69 | 78 | 67 | 105 | 53 | 5 | 52 | 32 |

both the UAG and MAG. High level of arousal was observed in three participants: two seniors from the UAG and one senior from the MAG (Table 7).

Simultaneous movements of both eyes in the same direction are called saccades. Typical characteristics of saccadic movements are 400° s for the maximum velocity, 20° for the

**Table 5  Mean and Standard deviations of ACC and GYRO signals.**

| Parameter | | Mean (SD) Acceleration | | | Mean (SD) Head rotation | | |
|---|---|---|---|---|---|---|---|
| | | **ACCX** | **ACCY** | **ACCZ** | **GYROX** | **GYROY** | **GYROZ** |
| MAG | Junior | −561.8 (221.29) | 1,227.6 (213.85) | −1,517 (175.97) | −5.88 (101.2) | 3.50 (91.83) | −11.54 (232.85) |
| | Mid-senior | −496.4 (369.56) | 1,119.8 (218.99) | −1,583 (199.87) | −4.18 (39.90) | 0.70 (39.66) | −12.53 (65.30) |
| | Senior | −457.09 (444.64) | 1,079.42 (613.60) | −1,478.72 (374.89) | −2.37 (153.49) | −2.30 (166.40) | −4.91 (332.81) |
| UAG | Junior | −620 (303.67) | 1,325.4 (304.95) | −1,361 (293.07) | −4.20 (106.12) | −0.19 (111.11) | −1.71 (247.03) |
| | Mid-senior | −492.8 (358.27) | 1,204.7 (278.72) | −1,506 (253.17) | −2.24 (97.78) | 5.95 (88.48) | −20.24 (221.69) |
| | Senior | −573.00 (366.77) | 1,307.44 (358.25) | −1,364.05 (291.74) | −1.14 (126.79) | 3.57 (162.21) | −7.81 (324.82) |

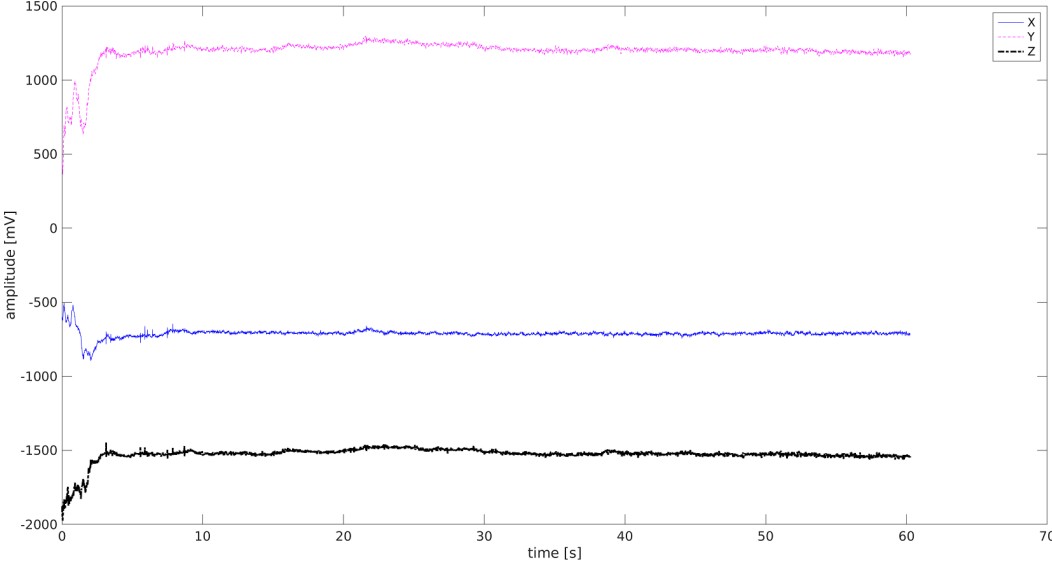

**Figure 8  ACC signal -Junior Student MAG.** X (blue) Y (pink) Z (black)-three spatial dimensions.

amplitude, and 80 ms for the duration. Fixations are static states of the eyes during which the gaze is held at a specific location (*Cognolato et al., 2017*). Humans typically alternate saccadic eye movements and fixations while perceiving their environment and the ACC signal and the EOG signal are needed to evaluate these motions.

The pilot scale of arousal was created to determine levels of stress, however, its usefulness should be confirmed in further research. There is an arbitrarily measurable difference in ACC and EOG when using JINS glasses and, inversely, eye movements can be tracked by analyzing the changes in the EOG signal. The electrode pairs capture the horizontal and the vertical components of eye motion.

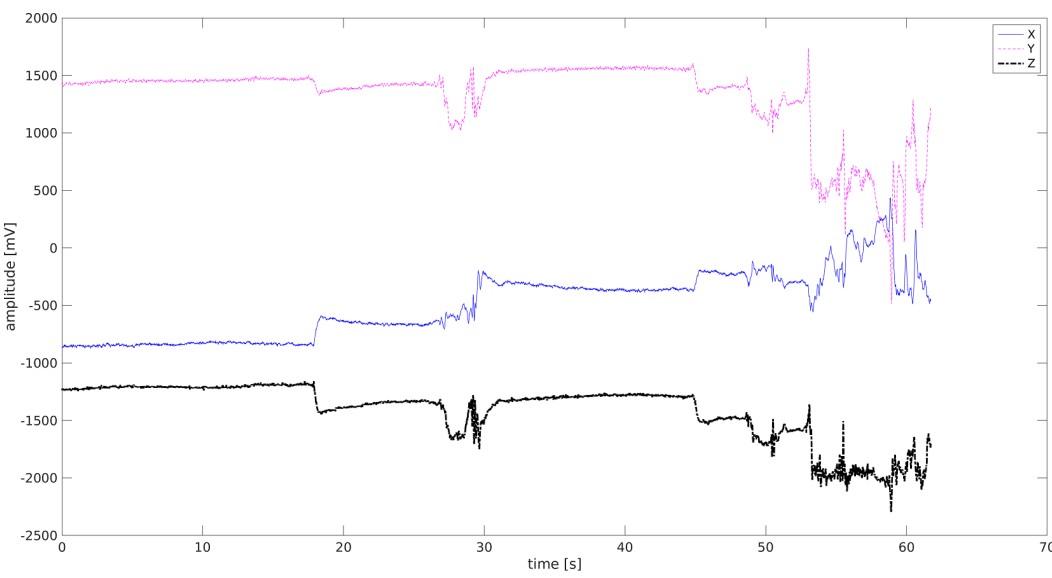

**Figure 9** **ACC signal-Junior Student UAG.** X (blue) Y (pink) Z (black)-three spatial dimensions.

**Table 6** Mean values and standard deviations (SD) of ACC_X signal of each study participant.

| Participant ID | 1 | 2 | 3 | 4 | 5 | 6 | 7 | 8 | 9 | 10 |
|---|---|---|---|---|---|---|---|---|---|---|
| Junior UAG | −229.2 (64.2) | −420.7 (48.5) | −457.6 (12.3) | −509.2 (87.9) | −496.3 (283.1) | −464.7 (185.4) | −843.3 (59.2) | −641.5 (112.9) | −757.5 (23.5) | −795.0 (126.7) |
| Junior MAG | −204.7 (285.3) | −667.9 (64.7) | −813.6 (116.8) | −737.3 (101.6) | −711.5 (26.6) | −600.2 (62.0) | −1051.0 (159.8) | −183.7 (225.8) | −474.5 (115.9) | −831.6 (59.9) |
| Mid Senior UAG | −383.2 (74.7) | −1271.6 (140.5) | −332.4 (102.2) | −575.1 (81.3) | −414.2 (108.5) | −874.5 (120.8) | −167.7 (43.1) | −674.3 (169.9) | −78.2 (51.9) | −160.8 (24.0) |
| Mid Senior MAG | −287.6 (65.8) | −1272.5 (195.3) | −483.0 (121.3) | −694.4 (179.0) | −446.5 (56.5) | −762.2 (311.3) | −190.1 (47.9) | −382.3 (152.6) | −303.5 (54.4) | −134.9 (204.2) |
| Senior UAG | −474.9 (95.0) | −979.8 (807.8) | −179.1 (34.6) | −445.9 (249.6) | −697.9 (368.0) | −499.9 (79.2) | 166.8 (5.5) | 435.2 (1,905.9) | −623.5 (201.5) | −1,021.4 (134.6) |
| Senior MAG | 167.8 (5.1) | −802.6 (454.6) | −324.3 (182.9) | −459.4 (152.3) | −963.7 (93.8) | −163.7 (95.1) | −1,049.5 (179.5) | −543.6 (212.9) | −461.6 (241.1) | −458.1 (258.4) |

**Table 7** Pilot scale of arousal-distribution of participants.

| | Junior (III) | | Mid-senior (IV) | | Senior (V) | | Total | |
|---|---|---|---|---|---|---|---|---|
| | UAG (n) | MAG (n) | UAG (n) | MAG (n) | UAG (n) | MAG (n) | UAG (n) | MAG (n) |
| Low arousal | 3 | 3 | 3 | 2 | 2 | 3 | 8 | 8 |
| Normal arousal | 7 | 7 | 7 | 8 | 6 | 6 | 20 | 21 |
| High arousal | 0 | 0 | 0 | 0 | 2 | 1 | 2 | 1 |

EOG requires good electrode placement on the horizontal and vertical axes of eye motion to prevent crosstalk between both components. Typically, the signal amplitudes resulting from horizontal eye movements are larger than those from vertical movements. Therefore, crosstalk affects the vertical component more severely. Signal crosstalk poses problems for the robust detection of eye movement events and eye gaze tracking for which both components need to be analyzed simultaneously. In the human eye, only a small central region of the retina, known as the fovea, is sensitive enough for most visual tasks. There are differences between the measurements of HR and gyration in a student who has knowledge of stress and stress-management techniques and a student who does not (*Bulling, Roggen & Tröster, 2008*).

The statistical tests and histograms obtained from data from the JINS glasses and pulse rates enabled us to systematize and determine a scale for the differences caused by stress.

### *Summary of the results*

We found that stress management training helped one subject from the senior group to progress from a high level of arousal to a low level of arousal and one subject from the mid-senior group moved from the low arousal level to the normal arousal level after training. According to the literature, arousal may be helpful in a high-pressure situation. Body signals resulting from stress, including butterflies in the stomach, increased heart rate, and muscle tension, are associated with an adaptive stress reaction and improved task performance (*Jamieson et al., 2016*). We found no direct relationship between eye movement and the definition of stress. However, we were able to demonstrate with JINS glasses that the training had an effect on eye movement. We also found that mid-senior and senior MAG students had a lower number of peaks, which may indicate the effectiveness of anti-stress training.

## DISCUSSION

According to the literature, dental students experience considerable stress and depressive symptoms during their theoretical and hands-on training (*Al Faris et al., 2016*; *Farokh-Gisour & Hatamvand, 2018*; *Chandrasekaran, Cugati & Kumaresan, 2014*; *Elani et al., 2014*). Dental schools are known for their highly demanding and stressful learning environments and high levels of depression among their students which is more pronounced in women (*Al Faris et al., 2016*).

The most common data collection method to evaluate stress levels are questionnaires, such as the Dental Environment Stress (DES) questionnaire (*Manolova et al., 2019*; *Babar et al., 2015*) or the Depression, Anxiety and Stress Scale (DASS-21) (*Basudan, Bnanzan & Alhassan, 2017*). We sought to determine the usefulness of JINS glasses for the assessment of stress levels among dental students when they perform dental procedures. The experiment was performed using phantom teeth to avoid patient contact and remove the stress variable of the subjects' awareness of working on live patients. The student's stress was related to the time restriction and the need to precisely perform the dental procedure. The usefulness of the tools (the smart glasses and the smartwatch) was assessed to measure objective symptoms and to control the vital parameters that confirmed anxiety and stress.

The JINS glasses have been used for the objective evaluation of physical parameters directly related to the intensification of stress (*Ogawa, Takahashi & Kawashima, 2016*). We used JINS glasses for the first time with dental school students during their performance of a task to evaluate and obtain data for analysis and further classification.

There are few reports in the literature on tools used for measuring and analyzing the objective responses of the body to increasing levels of stress and only one pilot study on the subject concerning the area of dentistry. *Hunasgi et al. (2018)* used the sensor and software located in smartphones to record and analyze the heart rate, oxygen saturation, and stress in dental students before and after their university practical exam.

Our own pilot study was conducted using a phantom clinical simulation. The least experienced students felt the highest level of stress while fifth year students who had worked with living patients managed their stress better. These results are consistent with the literature (*Piazza-Waggoner et al., 2003*; *Ali et al., 2018*; *Davidovich et al., 2015*), which indicates that clinical work with a living patient is more stressful than the first contact with mechanical dental equipment and the precise preparation of the cavity in a phantom tooth, despite having additional training or experience.

Our results are similar to those of other studies that also had a limited number of participants and data, and reveal that electronic tools with software and sensors can be used to detect stress levels at any point of time. The JINS glasses were calibrated in preclinical operating conditions, and the graphs were analyzed to preliminarily determine the readings that corresponded with a particular stage of concentration or stress. We intend to continue this study by assessing the stress levels of large groups of students performing dental procedures, including students practicing on phantom teeth and later on patients. We will compare the results to the patterns obtained in this preliminary experiment.

## CONCLUSION

Our pilot study shows that electronic tools, such as the JINS glasses with software and sensors, can measure different levels of stress experienced by dental students in preclinical operating conditions. We currently lack a clear definition of the states of stress. However, research using modern wearable devices can record baselines and establish patterns of stresses to allow for accurate assessments and management of stress in the future.

Future research should seek to determine whether the prevalence of severe depressive symptoms among students in different disciplines is due to the innate nature of the discipline, the type of curriculum being studied, or the forms of knowledge and manual skills required.

## ACKNOWLEDGEMENTS

The authors wish to thank all of the students who participated in this study.

### Funding
The authors received no funding for this work.

### Competing Interests
Marek Pachoński is an employee of Pachonscy Dental Clinic.

### Author Contributions

- Katarzyna Mocny-Pachońska conceived and designed the experiments, performed the experiments, analyzed the data, authored or reviewed drafts of the paper, and approved the final draft.
- Rafał Doniec conceived and designed the experiments, analyzed the data, authored or reviewed drafts of the paper, and approved the final draft.
- Agata Trzcionka performed the experiments, authored or reviewed drafts of the paper, and approved the final draft.
- Marek Pachoński conceived and designed the experiments, prepared figures and/or tables, and approved the final draft.
- Natalia Piaseczna, Szymon Sieciński, Oleksandra Osadcha and Patrycja Łanowy analyzed the data, prepared figures and/or tables, and approved the final draft.
- Marta Tanasiewicz conceived and designed the experiments, performed the experiments, authored or reviewed drafts of the paper, and approved the final draft.

### Human Ethics

The following information was supplied relating to ethical approvals (i.e., approving body and any reference numbers):

Consent for carrying out the experiments was issued by the Ethical Commission of Medical University of Silesia against resolution number KNW/0022/KB1/79/18 taken on October 16, 2018.

### Data Availability

The raw data are available in the Supplementary Files.

### Supplemental Information

Supplemental information for this article can be found online at http://dx.doi.org/10.7717/peerj.8981#supplemental-information.

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
