# Peer review of "Evaluating the stress-response of dental students to the dental school environment"

_PeerJ, doi:10.7717/peerj.8981_

## Round 0.1 · original submission · Major Revisions

Please pay particular attention on the comments related to far reaching conclusions based on previous literature, the statistic issues and overall formatting and grammar.

Reviewer 1 ·

Basic reporting

The grammar was generally fine, but I found a few unclear and awkwardly written sentences. For example,

In the introduction and Objectives section, “Results show that the heart rate of more experienced students had lower values than that of their the juniors.” Better, “Results show that the heart rate of more experienced students had was than that of their the juniors.” Also line 222

“…in further stages of multidirectional project.” Unclear what multidirectional project is. Also line 30

“…working with phantoms.” Should this be phantom teeth? Line 54

“… medical students experience the high levels of anxiety and…” Remove the word the. Line 64

“…have to master manual professional workshop…” Line 70 awkward

Intro should be revised, some redundancy Lines 71-72 is repeat of 64-66. 101-102 also similar

“Only one Senior student from UAG group was qualified to law activity compartment.” Line 276. What is “law activity compartment?”

The entire introduction, lines 53 – 112 is one paragraph.

“Thanks to this storage, which changes more dynamically and to a greater extent than the others.” Line 285. This is not a complete sentence and what is meant by “storage”?

“…software and sensors can be used to detect different level of activities in dental students…” Should this be stress-related activities? Line 341

The literature cited provided sufficient background, the structure of the article was appropriate, and the figures were relevant. I think the manuscript needs be edited for grammar.

Experimental design

The article is within the scope of PeerJ. The aim of the article was given as, “The main task of the researchers was to create an objective scale which could be implemented to determine the level of stress in further stages of multidirectional project.” I think goal would be a better word than task. It is the authors’ opinion that a method currently does not exist. It is unclear to me what a “multidirectional project” is?

It is my opinion that the study was ethical, and the materials section provided sufficient detail.

The authors repeat by saying, “The aim of this research was to find under medical simulation condition a way to standardize indicators useful for objective measurements of stress level.” Line 116. I am not sure this was accomplished. They thoroughly examined stress of students performing a dental simulation. However, I do not think they addressed how it was standardized. A number of measurements were taken, but did those measurements accurately measure stress?

A lot was written about the students’ results, but the results were not related to stress, the aim of the paper. This includes the marks given in Tables 1 and 2. Therefore, I do not think this is relevant. In addition, “Statistically important differences (t-student test) were obtained for the shape of prepared teeth” Line 206-9. Is this relevant?

“…differences were obtained in the shape of the phantom tooth worked upon by the junior,…” It might be my unfamiliarity with the simulation, but this statement is unclear to me. Students were given a phantom tooth to work on, so how is there a statistically significant difference in the phantom teeth? Was the difference in the printed teeth? Also lines 39-40. Again, I do not think this is related to the aim of the study.

Validity of the findings

I believe that the authors provided all data they used for their conclusions.

Table 5 head rotation data, especially for the junior class has a very large sd. Hard to draw any conclusions from these data.

In the Conclusion, the authors state, “The obtained results are significant and suggest that it would be useful to create an objective scale for stress measurement.” I do not see how the conclusion that it would be useful to create an objective scale for stress measurement was demonstrated. The statistical analyses demonstrated differences in the study groups, but how does that relate to the study aim? Also lines 44-45.

Additional comments

The stresses attributed to dental students apply to most students in the healthcare field.

In the introduction the authors write, “These findings support the results demonstrated by Abu-Ghazaleh et al. that stress in dental students at the University of Jordan increased over the years of the clinical training. Specifically, fifth-year students showed high levels of psychological stress due to awareness of expectations and responsibilities.” Then why did freshmen in this study have higher stress than seniors? Lines 86-88. Authors should comment on this.

“The negative impact of depression on dental students continues after graduation [13]. Does this mean that students with stress in dental school continue to have stress due to dental school, or is the stress due to different causes? Line 89.

When graduates begin their practice, is stress lower in graduates from programs with better clinical education? Lines 93-96

“Dental schools are known to have highly demanding and stressful learning environments and higher level of depression which often affects women [1] Higher compared to what? Line 305-6

Reviewer 2 ·

Basic reporting

Line 54: change “, already when” to “; from when”
Line 54: change “and start working with phantoms and” to “working with phantoms”
Line 59: add comma after diagnostics
Line 70: add comma after direct contact with the patient
Line 109-112: move to conclusion
Line 115: delete “to find under medical simulation condition a way”
Line 118: reword
Line 138 add comma before “and”
Line 152 add period
Line 169: check spacing before and after period
Line 173: check extra spaces
Line 175-179: explain in plain English and then add code in an appendix
Line 181: do you mean a bitmap file?
Line 184-186: please explain in plain English and add code in an appendix.
Line 201-202: delete sentence “Results are presented…”
Line 203-204: delete sentence “A value of p<0.05”
Line 294: check for extra spaces
Line 295: delete “It has been shown that”
Line 298-300: Reword sentence.
Line 327-330: Reword. “Both their preliminary results and ours show that electronic tools with software and sensors can be used to detect stress level at any point of time during a procedure.”
Table 3: Format vertically so the Table fits on the page.
Table 5: Please format vertically so the Table fits on the page.

Experimental design

Line 133: clarify how the students were selected, “randomly”, “from a class”, “volunteer”
Line 167 please provide reference for the algorithm, the C# code in an appendix, or github link
Line 201-204: please describe how you are adjusting for multiple testing
Line 206-208: please justify why you are using a t-test to compare pairwise means without adjusting for multiple comparisons.
Table 6: Please provide mean and standard deviation

Validity of the findings

Line 232-233: Since you are performing 60 experiments, your family wise Type I error is very high. Please explain how you are adjusting for multiple comparisons.

Figure 13: There is a lot of noise in this graph. Does this affect your analysis? How does this impact your research?


Table 4: Please provide more than just p-values

Table 5: What impact does the signal to noise ratio play in your analysis? Have you considered standardizing your variables for this analysis?

Additional comments

This paper has improved from the first time submitted, however there are still major revisions that need to be made. Mainly around the issue of multiple testing. This statistical issue needs to be addressed prior to publication. The methods section on statistical analysis should be expanded. In particular the authors note "p< 0.05" leads to rejection. My main concern is that statistics is more than just about p-values, it is about the actual effect size and the impact that would make on patients. This is not addressed in the paper. In addition, the authors should go through another round of editing for clarity, grammar, and presentation.

Reviewer 3 ·

Basic reporting

No comment.

Experimental design

No comment.

Validity of the findings

No comment.

Additional comments

The main concern about this paper is on the selection of the students. I understand that the authors randomly selected the students and the only criterion for inclusion in the study groups is being a student. Are there any exclusion criteria which the authors believe that may affect the gathered results?

The abbreviations should be written in full version in their first use.

---

## Round 0.2 · Minor Revisions

The quality of your manuscript has certainly improved compared to the previous version. However, additional improvements are needed before publication:

- Please read the manuscript carefully. There are multiple structural issues in paragraphs (e.g. double spacing between all words, multiple words underlined in blue suggesting Microsoft Word detects formatting issues). There are also grammatical errors. For example:
1) there is a verb missing in the results section of the abstract; page 4 starts a new paragraph with the word "it";
2) authors are misusing the word THE (e.g. page 5 "thee researchers", "the ANOVA")
3) using the word appliance (device?) to refer to smartwatches and glasses.

These examples were found doing just one quick read. There is no copy editing in PeerJ, therefore the article will not be ready for publication until all of these errors are corrected.

- The rebuttal states that the objectives have been clarified. I would have to disagree with this statement. The objective sentence should be specific, including the population studied. It is not clear what an arbitrary and measurable study means. This is not an objective. Finally, how and why stress simulation was interpreted as stress? there were major comments from the reviewers in the last edition about defining stress, and this has not been addressed. The goal was an arbitrary, measurable study (not needed) of the differences in the behaviour (measured how, variables?) of students *(what population?) given stress stimulation (defined as? variables?) and without it (define control further).

-The results are summarized as "Despite the small sample (meaning), this study demonstrated that anti-stress training (de-stress training not part of objectives/goals defined above) helped one person (one person significant?) from the senior group to progress from the low activity compartment to the normal activity compartment (what is activity? what is compartment?). Sentences describing results (and objectives above) need definition of what the authors are measuring and what those variables mean. Note that the reviewer asked "Only one Senior student from UAG group was qualified to law activity compartment." Line 276. What is "law activity compartment?" and you replied: "It should be low activity compartment", without defining what low compartment is.

-Many important reviewer questions have not been properly addressed. Examples include

1) Table 5 head rotation data, especially for the junior class has a very large sd. Hard to draw any conclusions from these data. This is a very good remark. Your answer: From these data we can conclude that the Juniors made much more head movements against the rest of students. Additional comment: the reviewer's point is that is the SD is too big, your conclusions are invalid, and no additional statistics are used in this answer to support your conclusion.

2) Line 206-208: please justify why you are using a t-test to compare pairwise means without adjusting for multiple comparisons. Answer: Statistical analysis has been improved. Additional comment: provide rationale.

3) There is a lot of noise in this graph. Does this affect your analysis? How does this impact your research- Graphs 13 and 14. Your response: Authors agree with the reviewer that it’s a lot of noise in these graphs. We are not able to show in paper all graphs, so we have decided to put them on one graph. Data were not analyzed from these graphs. These are the exemplary graphs. Additional comments: unclear why all graphs were put together in one graph at the detriment of clarity/quality if data was not analyzed from it. Graphs need to be improved. Consider using graphs where the data is analyzed rather than exemplified, and move examples to appendix. In addition, the noise effect is not answered in this reply.

4) Table 5: What impact does the signal to noise ratio play in your analysis? Have you considered standardizing your variables for this analysis? Your answer: Of course, the noise had an impact on the experiment.The experiment is on a small sample, we are looking for opposing classes.Standardization is not needed, but in subsequent analyzes with a large number of samples, it will certainly be necessary. Additional comments: the noise effect is not clarified, just confirmed. It's is still unclear why standardization is not needed.


-The method section needs further clarification for the audience to fully understand how the experiment was done, including an explanation of variables measured, compartments, anti-stress training. In addition, there are sentences in the results section that belong in the methods, together with an explanation, e.g. "The data collected during the first four seconds of the experiment (when the participant starts the procedure) and the last four seconds (when the procedure is completed) were not used for analysis, as the largest change in values were observed during that time. In total, 60 experiments were conducted. For proper interpretation of the results, one common episode that caused rapid rises of amplitude in a signal was chosen".

-Authors should limit conclusions to the evidence or literature. For example, "This behavior indicates that students from MAG group find the exercise a less comfortable situation. They feel more confident and have wider range of space perception" What is the evidence to support the statement that they feel more confident?
In addition, to make comments like this: " Typical characteristics of saccadic movements are 400°/s for the maximum velocity, 20° for the amplitude, and 80ms for the duration. Fixations are static states of the eyes during which the gaze is held at a specific location. Humans typically alternate saccadic eye movements and fixations while perceiving their environment, thus alternatives such as the ACC signal and the EOG signal are needed" citations should be added.
The following section also needs citations: "Inversely, eye movements can be tracked by analyzing these changes in the EOG signal. The electrode pairs capture the horizontal and the vertical component of eye motion.EOG requires good electrode placement, i. e. on the eyes' horizontal and vertical axes of motion, as otherwise increased crosstalk between both components occurs. Typically, the signal amplitudes resulting from horizontal eye movements are larger than those from vertical movements. Therefore, crosstalk affects the vertical component more severely. Signal crosstalk poses problems on the robust detection of eye movement events and eye gaze tracking for which both components need to be analyzed simultaneously. In the human eye, only a small central region of the retina, the fovea, is sensitive enough for most visual tasks, but HR and gyration between a student who has knowledge of stress and how to neutralize it, and a student who does not have such knowledge."

All in all, please read the manuscript carefully for structure, grammar and clarity. Clarity is specially needed in the method section to explain fully all the aspects of the experiment the authors refer to in results. Consider re-doing graphs and add citations to support statements of fact.

---

## Round 0.3 · accepted · Accept

Dear authors, please adjust the comments I made in the document.